# Evidence of Long-Lasting Humoral and Cellular Immunity against SARS-CoV-2 Even in Elderly COVID-19 Convalescents Showing a Mild to Moderate Disease Progression

**DOI:** 10.3390/life11080805

**Published:** 2021-08-09

**Authors:** Bastian Fischer, Christopher Lindenkamp, Christoph Lichtenberg, Ingvild Birschmann, Cornelius Knabbe, Doris Hendig

**Affiliations:** Institut für Laboratoriums-und Transfusionsmedizin, Herz-und Diabeteszentrum Nordrhein-Westfalen, Universitätsklinik der Ruhr-Universität Bochum, Georgstrasse 11, 32545 Bad Oeynhausen, Germany; clindenkamp@hdz-nrw.de (C.L.); clichtenberg@hdz-nrw.de (C.L.); ibirschmann@hdz-nrw.de (I.B.); CKnabbe@hdz-nrw.de (C.K.); dhendig@hdz-nrw.de (D.H.)

**Keywords:** SARS-CoV-2, COVID-19, cellular and humoral immunity, NK-cells

## Abstract

We here evaluate the humoral and cellular immune response against SARS-CoV-2 in 41 COVID-19 convalescents. As previous studies mostly included younger individuals, one advantage of our study is the comparatively high mean age of the convalescents included in the cohort considered (54 ± 8.4 years). While anti-SARS-CoV-2 antibodies were still detectable in 95% of convalescents up to 8 months post infection, an antibody-decay over time was generally observed in most donors. Using a multiplex assay, our data additionally reveal that most convalescents exhibit a broad humoral immunity against different viral epitopes. We demonstrate by flow cytometry that convalescent donors show a significantly elevated number of natural killer cells when compared to healthy controls, while no differences were found concerning other leucocyte subpopulations. We detected a specific long-lasting cellular immune response in convalescents by stimulating immune cells with SARS-CoV-2-specific peptides, covering domains of the viral spike, membrane and nucleocapsid protein, and measuring interferon-γ (IFN-γ) release thereafter. We modified a commercially available ELISA assay for IFN-γ determination in whole-blood specimens of COVID-19 convalescents. One advantage of this assay is that it does not require special equipment and can, thus, be performed in any standard laboratory. In conclusion, our study adds knowledge regarding the persistence of immunity of convalescents suffering from mild to moderate COVID-19. Moreover, our study provides a set of simple methods to characterize and confirm experienced COVID-19.

## 1. Introduction

The first cases of a novel respiratory disease occurred in Wuhan, China, in late December 2019. Polymerase chain reaction (PCR) testing identified the novel coronavirus severe acute respiratory syndrome coronavirus type 2 (SARS-CoV-2) early as a causative agent of the disease, which has been consequently named coronavirus disease 2019 (COVID-19) [1]. The virus has since spread worldwide and was classified as a pandemic by the WHO on 11 March 2020 [2]. While the first laboratory-confirmed case occurred in Germany on 24 January 2020 [3], the virus has so far infected millions of people worldwide. Two other coronaviruses with an increased pathogenic potential for humans have appeared in the last two decades: Severe acute respiratory syndrome coronavirus type 1 (SARS-CoV-1) occurred between November 2002 and June 2003 (8000 cases, 776 deaths) and Middle East respiratory syndrome coronavirus (MERS-CoV) in 2012 spread mainly on the Arabian Peninsula and infected 2428 people, 838 of whom died [4]. SARS-CoV-2 shows a sequence homology to SARS-CoV-1 of about 79.5%, whereas only a homology of 50% is found compared to MERS-CoV [5]. SARS-CoV-2 is an enveloped, single-stranded RNA virus, which is composed of four structural proteins: Spike (S), nucleocapsid (N), membrane (M) and envelope (E). Along with M and N, the E protein is responsible for the initiation and assembly of virus-like particles. Viral infection is realized by the binding of the trimeric S protein to the host cell’s angiotensin-converting enzyme 2 (ACE2) receptor. Thereby, the ectodomain of the S protein, which consists of the S1 subunit (containing the receptor binding domain: RBD) and the membrane fusion subunit (S2), plays a superior role [6]. The vast majority of infections are mild or even asymptomatic; however, the infection fatality rate is around 0.5–1%, with the probability of fatal outcomes increasing with age [7]. The knowledge about a possible post-infection immunity is still limited, and most studies include data from patients with more severe courses. It is assumed that both the humoral and cellular immune response have an impact on the severity. Most convalescents show detectable anti-SARS-CoV-2 IgG antibody levels between 10 and 21 d after infection [8]. However, there is evidence that some people, primarily those showing a mild progression, exhibit a delayed humoral response or even show no seroconversion at all [9]. According to recent studies, antibody persistence appears to depend primarily on the antibody class and COVID-19 severity. Anti-SARS-CoV-2 IgA and IgM antibody levels seem to drop rapidly, but IgG antibodies against the virus are detectable for several months in patients with a moderate or severe course [10,11]. Furthermore, Wajnberg et al. found that more than 90% of people who show general seroconversion also express neutralizing antibodies that can be detected for months and are directed primarily against the viral S protein [12]. Some publications suggest that the cellular immune response also plays an important role concerning SARS-CoV-2 containment and COVID-19 severity. Several studies show that a specific T-cell response in people with mild disease progression is triggered shortly after infection [13,14]. By contrast, people with a severe course of the disease are more likely to show a dysregulated cellular immune response [15,16]. Zuo et al. were able to demonstrate a robust T-cell response six months post-infection in all 100 initial seropositive individuals included in their study [17].

In our study, we characterized the cellular immune response of German convalescent blood donors suffering from mild to moderate COVID-19 compared to healthy blood donors without any history of SARS-CoV-2 infection. For this purpose, we modified a commercial enzyme-linked immunosorbent assay (ELISA) to rapidly and reliably measure the IFN-γ expression of leucocytes stimulated with a SARS-CoV-2-specific peptide pool covering different viral proteins. We examined the amounts of anti-SARS-CoV-2 IgA and IgG antibodies of convalescents over a period of up to eight months to assess the persistence of humoral immunity. We also provided detailed information on the profile of antibodies directed against different antigens of the new coronavirus using a novel multiplex bead-based assay.

## 2. Materials and Methods

### 2.1. Human Donors

The convalescents included in our study had a mild or moderate disease course not requiring hospitalization, and SARS-CoV-2 RNA was initially detected by PCR. Blood samples of patients recovered from COVID-19 (*n* = 41; age: 54 ± 8.4; male: 57%; female: 43%) were obtained between 28 and 228 d after the onset of symptoms. An average of nine plasma donations were collected from each convalescent individual (range: 2–21 donations) during the study period. Healthy regular blood donors (*n* = 18; age 38 ± 13; male: 42%; female; 58%) with no known past or present COVID-19 disease served as controls. All probands of this study showed no signs of disease, such as fever or increased leukocyte counts, because potential blood donors with any suspicion of infection are excluded from donating blood. Characteristics of all donors are listed in Appendix A.

All donors underwent a medical examination before donation. Samples were collected in accordance with the German Act on Medical Devices for the collection of human residual material. Ethical approval was obtained from the ethical committee of the HDZ NRW in Bad Oeynhausen (Reg.-No. 670/2020).

### 2.2. Stimulation of Immune Cells Using the SARS-CoV-2 Peptide Pool

An amount of 1 mL heparinized whole blood was treated with a SARS-CoV-2-specific peptide pool (Miltenyi Biotec, Bergisch-Gladbach, Germany) for immune cell stimulation. The peptide pool contained synthetic peptides whose sequences derived from the viral S, N and M proteins (final concentration of each peptide: 1 µg/mL). Treatment of whole blood with water served as a negative control. Stimulation with QuantiFERON Monitor (QFM) LyoSpheres (Qiagen, Hilden, Germany), which stimulate and activate different cell types of the innate and adaptive immune system, was used as a positive control. Samples were mixed by inverting and immediately incubated at 37 °C for 20–24 h. After incubation, samples were centrifuged (10 min; 2000× *g*), and plasma was collected and stored at −20 °C until use.

### 2.3. Determination of Interferon-γ in Plasma

The QuantiFERON Monitor ELISA was used (Qiagen, Hilden, Germany) for the analysis of the IFN-γ amount in the plasma of controls and convalescents. Both peptide-stimulated and unstimulated samples (negative control) were measured undiluted. The experimental procedure was conducted following the manufacturer’s instructions.

### 2.4. Determination of Anti-SARS-CoV-2 IgG and IgA Antibodies (Euroimmun)

Two commercial ELISAs (Euroimmun, Lübeck, Germany) were used for the determination of anti-SARS-CoV-2 antibodies (IgG and IgA). The assays were conducted automatedly using the Euroimmun Analyzer I system, following the provider’s instructions. Values are semi-quantitively expressed as ratios.

### 2.5. Measurement of Anti-SARS-CoV-2 S1/S2 IgG Antibodies (DiaSorin)

The LIAISON SARS-CoV-2 IgG chemiluminescent assay, designed to measure IgG antibodies against the S1 and S2 subunits of the viral S protein, was automatedly conducted using an XY device. Anti-SARS-CoV-2 IgG levels were quantitively expressed as arbitrary units (AU/mL) according to a standard curve.

### 2.6. Multiplex Assay for the Differentiated Detection of Anti-SARS-CoV-2 Antibodies

The LABScreen COVID Plus multiplex assay (ThermoFisher, Waltham, MA, USA) was used to determine the expression pattern of anti-SARS-CoV-2 antibodies directed against differential viral epitopes (within the S, S1, S2, RBD and N protein). The assay was performed according to the manufacturer’s instructions.

### 2.7. Determination of Neutralizing Antibodies against SARS-CoV-2

The determination of neutralizing antibodies against SARS-CoV-2 was applied by using the cPass^TM^ SARS-CoV-2 Neutralization Antibody Detection KIT (GenScript, Piscataway Township, NJ, USA). Experimental procedure was performed following the manufacturer’s instruction using EDTA plasma. According to the manufacturer, values >20% were considered positive concerning neutralizing antibodies.

### 2.8. Flow Cytometry

Whole EDTA blood was analyzed by flow cytometry to determine the lymphocyte subpopulations and immune status of donor blood. Consequently, samples were incubated with a mixture of fluorochrome-tagged monoclonal antibodies (BD Multitest^TM^ 6-Color TBNK; Table 1) for 15 min at RT in the dark. Afterwards, each sample was mixed with 500 µL FACS Lysing solution (BD Biosciences, Franklin Lakes, NJ, USA) to remove erythrocytes and again incubated for another 15 min at RT in the dark. After incubation, samples were analyzed using a BD FACS Canto^TM^ II device. A total of 3000 individual cells were recorded for each sample using a dot plot combination of low angle forward scattered (FCS) and right angle scattered (SSC) laser light. Data were analyzed using BD FACS DIVA software.

A Sysmex XN-1000 (Sysmex, Kōbe, Japan) flow cytometer was used to determine white blood cells, lymphocytes and monocytes in donors’ EDTA blood.

### 2.9. Statistical Analysis 

Data are shown as mean ± standard error (SEM). The software GraphPad Prism 7.0 was used for statistical analysis of data. The non-parametric two-tailed Mann–Whitney U test was performed for statistical analysis. *p*-values of 0.05 or less were considered statistically significant.

## 3. Results

### 3.1. Increased IFN-γ Release in COVID-19 Convalescents after Peptide Stimulation 

Figure 1 shows the IFN-γ release of the unstimulated and stimulated whole blood of convalescent donors and healthy controls. Treatment with specific SARS-CoV-2 peptides led to a highly significant increased IFN-γ-release (3.93 ± 0.71 IU/mL vs. 0.34 ± 0.10 IU/mL; *p* < 0.0001) in convalescents compared to unstimulated whole blood. The release of IFN-γ did not differ significantly between the stimulated and unstimulated whole blood of healthy controls. There was no significant difference in the IFN-γ secretion between the unstimulated whole blood of convalescents and controls. The IFN-γ release in the whole blood of stimulated convalescents was significantly increased compared to the stimulated plasma of control donors (3.39 ± 0.71 vs. 0.43 ± 0.14; *p* < 0.0001).

### 3.2. No Correlation between IFN-γ Concentration and Anti-SARS-CoV-2 IgG Expression

The IFN-γ concentration was correlated to the anti-SARS-CoV-2 IgG expression. The expression of anti-SARS-CoV2 IgG was determined by two different methods: a semiquantitative ELISA assay (Euroimmun) and a quantitative CLIA test (DiaSorin). There was no significant difference in the IFN-γ release between the stimulated whole blood of convalescent donors showing low (IgG ratio 1.0–1.9), medium (IgG ratio 2.0–3.9) or high (IgG ratio >4) anti-SARS-CoV-2 IgG antibody ratios using the Euroimmun assay, as seen in Figure 2A. Additionally, no significant differences in the IFN-γ release were detected between grouped (low: 0–30 AU/mL; medium: 31–100 AU/mL; high: >100 AU/mL) anti-SARS-CoV-2 IgG concentrations in the quantitative DiaSorin assay (Figure 2C). These observations were confirmed by linear regression, which showed only a weak correlation between the IFN-γ concentration and anti-SARS-CoV-2 IgG expressions determined using the Euroimmun (r = 0.2831, Figure 2B) and DiaSorin (r = 0.2578, Figure 2D) assay, respectively.

### 3.3. Multiplex Assay for the Qualitative Detection of Anti-SARS-CoV-2 IgG Antibodies

The COVID-Plus multiplex assay was conducted to detect antibodies qualitatively against the S, S1, RBD, S2 and N of SARS-CoV-2. Antibodies against the viral S, S1, RBD and S2 protein were detected in all but one convalescent donors, as can be seen in Figure 3. Only 61% (25/41) of those who recovered from COVID-19 expressed antibodies against the viral N. While all controls were generally negative for the expression of SARS-CoV-2 specific antibodies as expected, antibodies to the viral S protein were detected in one control donor.

### 3.4. Expression of Neutralizing Anti-SARS-CoV-2 Antibodies

The expression of neutralizing antibodies against SARS-CoV-2 was increased significantly in the plasma of convalescent donors (60.82 ± 3.15% inhibition capability), as shown in Figure 4A. As might be expected, the inhibition capability in healthy controls was below the manufacturer’s cutoff of 20% (12.23 ± 1.02%).

In addition, a moderate correlation between the neutralizing and general anti-SARS-CoV-2 IgG antibody expression, determined by using the Euroimmun (Figure 4B,C, r = 0.7526) and DiaSorin (Figure 4D,E, r = 0.778) assay, was detected. In detail, a significant increase of inhibiting neutralizing antibodies was shown in the plasma of convalescent donors with medium (Euroimmun: ratio 2.0–3.9; DiaSorin: 31–100 AU/mL) in comparison to those with low (Euroimmun: ratio 1.0–1.9; DiaSorin: 0–30 AU/mL) IgG antibody levels (65 ± 3.96% and 67.42 ± 3.70% vs. 46.48 ± 3.65% and 46.55 ± 3.54%, respectively). Convalescents showing high (Euroimmun: ratio > 4; DiaSorin: >100 AU/mL) anti-SARS-CoV-2 IgG values in both assays tested also showed the highest inhibition capabilities (84.36 ± 3.51% and 93.91 ± 0.53%, respectively).

### 3.5. Higher Amount of Natural Killer Cells in Blood of Convalescent Donors

The composition of leucocytes in whole blood specimens of convalescents and healthy controls was analyzed by flow cytometry. There was no significant difference in the amount of immunological cell subpopulations between both groups, as shown in Figure 4, except of natural killer (NK) cells (Figure 5D). When compared to controls (7.68 ± 1.03), the NK cell expression was increased significantly in COVID-19 convalescents (10.76 ± 0.80%).

## 4. Discussion

The COVID-19 pandemic has claimed millions of lives so far worldwide. The first vaccines have been developed and approved in record time; however, many questions regarding the immunity of convalescents to the SARS-CoV-2 virus remain unresolved. In this regard, most studies focus on the humoral immune response, although cellular immunity also appears to have an impact on the severity of the disease. We stimulated whole blood with a SARS-CoV-2-specific peptide pool (including sequences of the viral S, M and N protein) and subsequently determined IFN-γ release to evaluate the cellular immune response of convalescent and healthy control donors. To the best of our knowledge, we are the first to use a modified version of the QuantiFERON Monitor assay (Qiagen, Hilden, Germany) for this purpose. The assay is particularly suitable if, as in our study, a general statement about IFN-γ release before and after stimulation is made. In order to be able to make cell-specific assessments regarding the release of IFN-γ, exemplarily commercial ELISPOT systems or special flow cytometry-based assays should be used. Nevertheless, the ELISA-based assay used in our study shows important advantages concerning its rapid and simple feasibility, its commercial ready-to-use availability (no manual plate-coating required) and the fact that a standard microplate-reader and no special software are required for the analysis. This allows the assay to be performed by any laboratory with standard equipment. Cells in our study were stimulated in a parallel setup using LyoSpheres pellets containing a CD3 T-cell receptor agonist and a viral TLR 7/8 ligand (resiquimod or R848) [18], provided by the manufacturer, for validation purposes and as a positive control. Cells of the control and convalescent blood donors were stimulable to a similar extent when using LyoSpheres (Appendix A). As the assay was originally designed and optimized for the use of LyoSpheres and the binding of the containing ligands leads to a very strong cellular immune response, we detected a much higher IFN-γ release when compared to the stimulation with the specific SARS-CoV-2 peptides. However, as this had no direct impact on the significance of our results, there is no need to use LyoSpheres in similarly designed studies in the future.

While the immune cells of convalescent donors secreted increased amounts of IFN-γ after stimulation with SARS-CoV-2 specific peptides, the cells of control donors did not respond significantly to the specific SARS-CoV-2 peptide pool at all. This is in accordance with data published previously and, therefore, confirms the validity of the assay. Le Bert et al. showed an increased IFN-γ release after peptide stimulation in 36 individuals tested, who all recovered from COVID-19. In contrast to our study, the authors isolated peripheral blood mononuclear cells (PBMCs) and stimulated them with synthetic peptides that covered only the viral N protein. The measurement of IFN-γ release was performed with the ELISpot system mentioned above [19].

Our data suggest that cellular memory to SARS-CoV-2 is stably maintained for at least eight months after the COVID-19 disease and that the amount of IFN-γ release after peptide stimulation does not seem to correlate to the time post symptom onset. These results substantiate previous findings from other research groups that identified specific memory B [20] and T [17] cells eight and six months, respectively, after SARS-CoV-2 infection. We further observed no correlation between the cellular and humoral immune responses of convalescent donors. In detail, neither semiquantitative (Euroimmun, r = 0.2831) nor quantitative (DiaSorin, r = 0.2578) anti-SARS-CoV-2 IgG assay results correlated with the respective immune cell IFN-γ release after stimulation with SARS-CoV-2-specific peptides. All convalescents were seropositive at first donation; however, a successive antibody decay was observed for most donors. Previous studies had already suggested an early decay or even seroreversion of IgG antibodies in individuals with a mild disease progression [21,22]. Nevertheless, IgG antibodies were still detectable in 95% of all donors (39/41) included in our study on the day of the last donation and, therefore, up to eight months after symptom onset. No or only equivocal anti-SARS-CoV-2 IgA antibodies were detectable in some convalescents (8/41, 20%) on the day of the last donation, whereby some were already seronegative at first donation (6/41, 15%). This accords with the results of previous studies, which postulated a comparably fast decay of this antibody class after SARS-CoV-2 infection [10,11]. However, an increased rate of false–positive results must be assumed due to the comparatively low specificity (71.5% [23]) postulated for the Euroimmun anti-SARS-CoV-2 IgA assay used.

We further conducted a bead-based multiplex assay to determine more precisely against which viral antigens the anti-SARS-CoV-2 antibodies are targeted. We were able to elicit that all except one convalescent proband formed antibodies against each of the S domains examined (S1, S2 and RBD domain). This is important information because it suggests broad-spectrum antibody expression, which increases the potential for at least partial antibody binding to occur in spite of viral mutations. While nearly all convalescent donors formed antibodies against the S protein, only 62.5% expressed antibodies against the viral N. Although data for the new coronavirus are lacking, different studies suggest that people infected with the SARS-CoV virus that appeared in 2004 were almost all positive for the expression of N protein-specific antibodies, whereas only about half of the convalescents expressed S protein-specific antibodies [24,25,26]. It is quite possible that the convalescents included in our study, who are apparently negative for N protein-specific antibodies, in fact, express antibodies that are not directed against the N-specific antigens coated on the microbeads of the assay used. Furthermore, the expression of antibodies against the viral N protein correlated well with the anti-SARS-CoV-2 IgG ratios determined in the serological Euroimmun assay (which detects antibodies against the viral S protein). This suggests a simultaneous degradation of N and S antibodies, whereby S antibodies may be detectable longer due to a higher initial titer.

As expected, no anti-SARS-CoV-2 antibodies were detected for most donors of the control cohort; however, antibodies against the viral S protein were identified for one control using the multiplex assay. This result initially suggested cross-reactivity, as no antibodies were detected for the residual targets. This was confirmed, as the immunological cells of this donor were not stimulable to release IFN-γ with the specific SARS-CoV-2 peptides. It is notable that the donor was also weakly positive in the remaining two anti-SARS-CoV-2 IgG serological assays, which are both designed for the detection of antibodies directed against the viral S protein. False-positive results may be due to the specificity of both serologic assays evaluated, being less than 100% [27]. It is interesting that the blood cells of the only convalescent donor for whom no antibodies against the S1, S2 and RBD domain were detectable in the multiplex assay were stimulable by SARS-CoV-2-specific peptides. For this donor, only equivocal seropositive IgG levels were detected in the two remaining seroassays. These examples suggest that especially SARS-CoV-2 infections that are longer past can be identified most reliably by measuring of IFN-y release after immune cell stimulation. To substantiate this, the main focus in follow-up studies should be on characterizing the cellular immunity of convalescents who initially expressed no or few antibodies or showed a rapid antibody decay.

Interestingly, neutralizing antibodies were detectable in all convalescent donors, whereas levels in all controls were below the cutoff, as expected. The values of the neutralizing antibodies correlated well with the quantitative (r = 0.7778) and semiquantitative (r = 0.7526) anti-SARS-CoV-2 IgG results obtained in the DiaSorin and Euroimmun assays, respectively. These results match those of Wajnberg et al., who detected neutralizing antibodies in a large proportion (>90%) of 30,082 individuals tested with mild to moderate disease progression. Neutralizing antibodies were thereby also persistent for months, and neutralizing antibody levels also correlated well with the anti-SARS-CoV-2 IgG antibody results determined [12].

We next characterized immunological cell subpopulations of control and convalescent donors. The only difference we observed concerned the occurrence of NK cells, which was increased significantly in convalescents. This is a very interesting finding, as previous studies suggested that NK cells tend to be decreased in COVID-19 patients [28,29,30]. Most of these were patients with a severe course of COVID-19, whereas our study involves individuals with mild to moderate COVID-19 disease. Decreased NK cell levels in severe COVID-19 cases are associated with insufficient degradation of infected cells, which, inter alia, results in elevated inflammation [31]. Increased NK cell counts in individuals with mild progression could probably, therefore, contribute to avert a “cytokine storm,” that is known to trigger a severe COVID-19 outcome [32]. Characterization of leucocyte subpopulations in donor blood was performed without stimulation with anti-SARS-CoV-2-specific peptides. Because NK cells are IFN-γ-secreting [33] immunological memory cells [34,35], the increased IFN-γ release of immune cells in the blood of convalescents after peptide stimulation could also be explained by an inductive formation of NK cells. However, this assumption needs further research on the involvement of NK cells in COVID-19 pathogenesis.

## 5. Conclusions

In conclusion, the cellular immune response in convalescent COVID-19 patients was determined by an adapted commercial assay in a comparatively easy and fast way. Our data show that the newly developed assay identifies convalescent individuals reliably and is superior in its specificity to the serological assays used in this study. The cellular SARS-CoV-2 -specific IFN-γ release described in our study might be helpful for the interpretation of the specificity of anti-SARS-CoV-2 antibodies detected, especially when experienced COVID-19 disease is suspected but PCR-confirmed SARS-CoV-2 infection is missing. In this study, the examined cohort includes convalescents with a mild disease course and a comparatively high mean age of 54 ± 8.4 years. Our results, therefore, also assume a long-lasting immunity against SARS-CoV2 infection in the elderly.

## Figures and Tables

**Figure 1 life-11-00805-f001:**
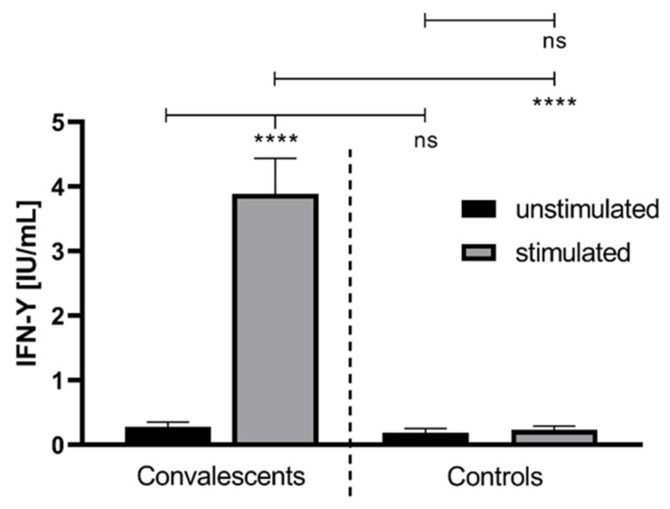
The IFN-γ concentration in unstimulated and stimulated whole blood of convalescent COVID-19 donors (n = 41) and healthy controls (n = 18). The IFN-γ release was monitored after treatment of the whole blood, donated from COVID-19 convalescents and healthy controls, with a SARS-CoV-2-specific peptide pool (grey bars). The latter contained synthetic peptides whose sequences derived from the viral spike (S), nucleocapsid (N) and membrane (M) proteins (final concentration of each peptide: 1 µg/mL). The treatment of whole blood with water served as a negative control (black bars). ****: *p* < 0.0001; ns: not significant (Mann–Whitney U test).

**Figure 2 life-11-00805-f002:**
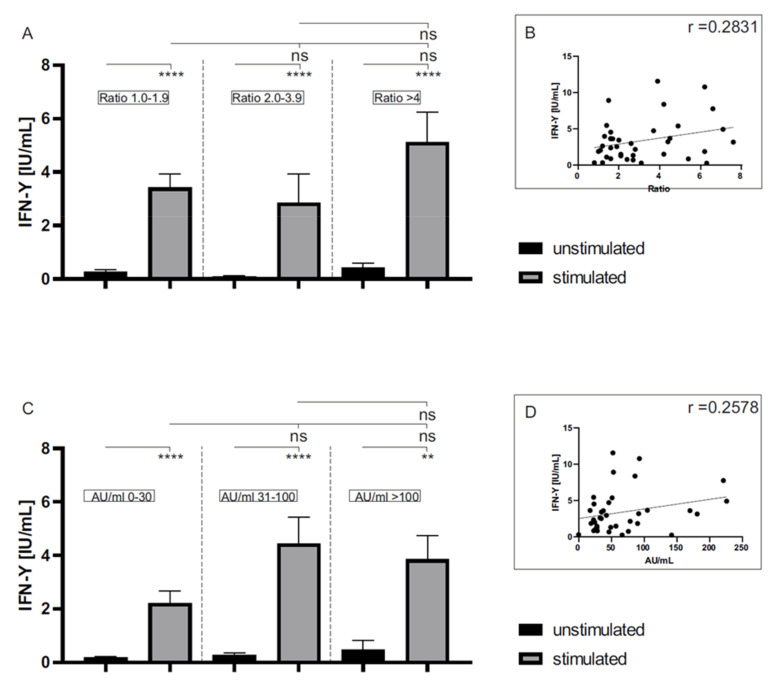
The IFN-γ release in unstimulated and stimulated whole blood of COVID-19 convalescent donors (n = 41) in relation to anti-SARS-CoV-2 IgG levels determined using two different serological assays. Anti-SARS-CoV-2 antibodies of convalescent donors, determined by the semiquantitative Euroimmun (**A**) or the quantitative DiaSorin (**C**) assay, were grouped and related to the IFN-γ release of unstimulated (black bars) and stimulated (grey bars) whole blood. A pool of synthetic peptides whose sequences derived from the viral S, N and M proteins (final concentration of each peptide: 1 µg/mL) was used for stimulation. Water treatment served as a negative control (unstimulated). In addition, the individual results of the antibody measurements were plotted against the respective IFN-γ release to perform a linear regression (**B**,**D**). ****: *p* < 0.0001; **: *p* < 0.002; ns: not significant (Mann–Whitney U test). (**D**) DiaSorin (r = 0.2578) assay.

**Figure 3 life-11-00805-f003:**
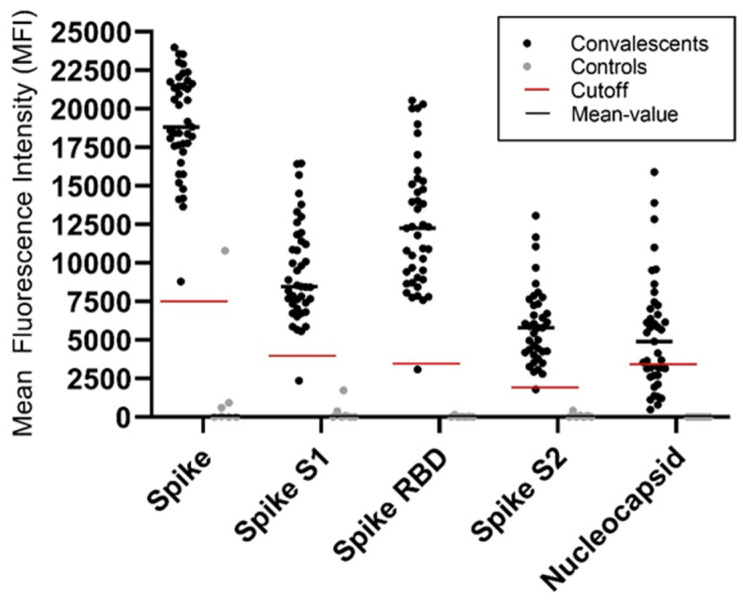
Qualitative detection of anti-SARS-CoV-2 antibodies in convalescents (n = 41) and healthy controls (n = 7) using a multiplex flow cytometry assay. The serum of convalescents and controls was used to identify antibodies binding to beads coated with various purified SARS-CoV-2 antigens. Samples were considered positive if they had a mean fluorescence intensity (MFI) value above the manufacturer’s cutoff. The individual cutoff for each antibody was set as follows: S: 7500; S1: 4000; RBD: 3500; S2: 1900; N: 3500.

**Figure 4 life-11-00805-f004:**
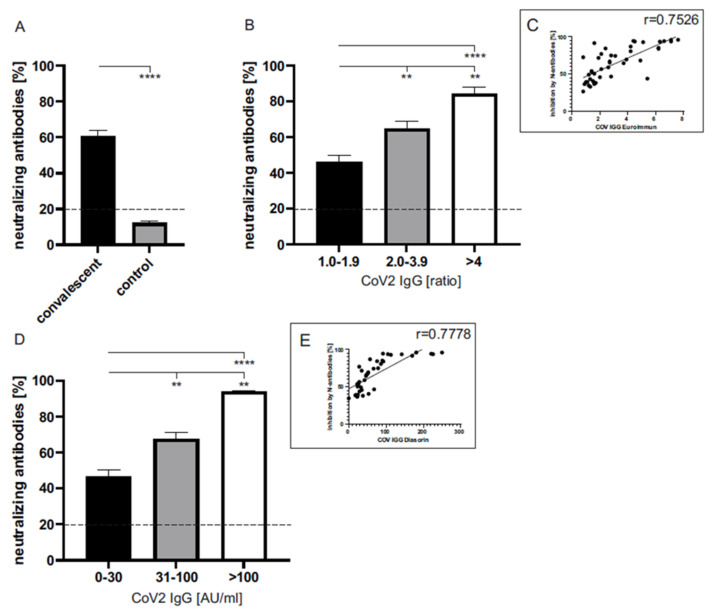
The inhibition capability of neutralizing antibodies in the plasma of convalescent COVID-19 donors (n = 41) and controls (n = 18). (**A**) The general inhibition capability of neutralizing antibodies in the plasma of convalescents (black bar) and healthy controls (grey bar). According to the manufacturer, the cutoff was set to 20%. (**B**) Correlation between the inhibition capability of neutralizing antibodies and anti-SARS-CoV-2 IgG ratios (Euroimmun) in the plasma from convalescent donors. Expressions were grouped as follows: low (IgG ratio: 1.0–1.9), medium (IgG ratio: 2.0–3.9) and high (IgG ratio: >4) (**B**,**C**). Correlation between the inhibition capability of neutralizing antibodies and anti-SARS-CoV-2 IgG concentrations (DiaSorin) in the plasma from convalescent donors. Expressions were grouped as follows: low (0–30 AU/mL), medium (31–100 AU/mL) and high (>100 AU/mL) (**D**,**E**). The dashed line symbolizes the manufacturer’s cutoff (<20%: negative; >20% positive). ****: *p* < 0.0001; **: *p* < 0.002 (Mann–Whitney U test).

**Figure 5 life-11-00805-f005:**
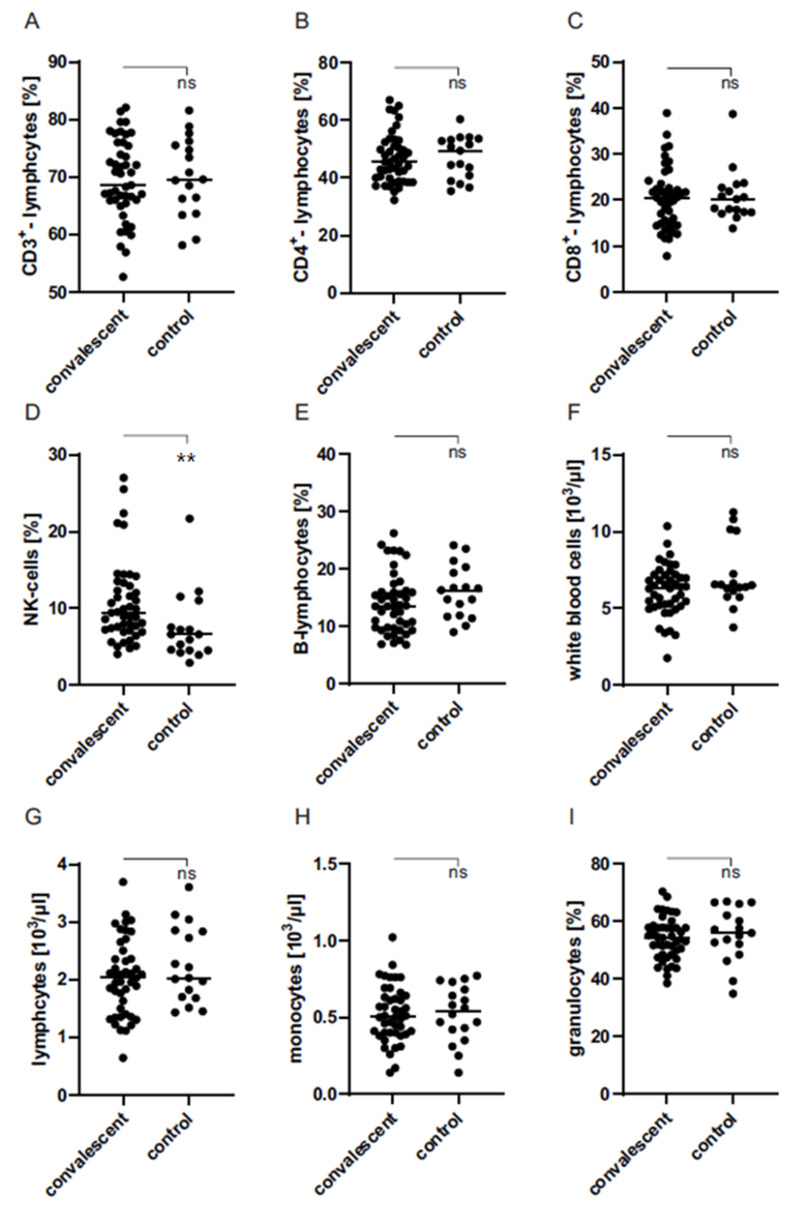
Determination of immunological cell subpopulations in whole blood specimens of COVID-19 convalescents (n = 41) and healthy controls (n = 18) by flow cytometry. Subpopulations: CD3+ lymphocytes (**A**), CD4+ lymphocytes (**B**), CD8+ lymphocytes (**C**), NK cells (**D**), B lymphocytes (**E**), white blood cells (**F**), lymphocytes (**G**), monocytes (**H**), granulocytes (**I**). **: *p* < 0.002; ns: not significant (Mann-Whitney U test).

**Table 1 life-11-00805-t001:** Antibodies used for the analysis of lymphocyte subpopulations.

Antibody	Marker	Manufacturer	Concentration (µg/mL)
Anti-CD3	FITC	BD Bioscience	2.3
Anti-CD4	PE-Cy^TM^7	BD Bioscience	1.5
Anti-CD8	APC-Cy^TM^7	BD Bioscience	6.3
Anti-CD16	PE	BD Bioscience	1.65
Anti-CD56	PE	BD Bioscience	1.1
Anti-CD19	APC	BD Bioscience	2.3

## Data Availability

Not applicable.

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
