# Peer review of "Evidence of Long-Lasting Humoral and Cellular Immunity against SARS-CoV-2 Even in Elderly COVID-19 Convalescents Showing a Mild to Moderate Disease Progression"

_life, 2021, doi:10.3390/life11080805_

Round 1

Reviewer 1 Report

The manuscript concerns a very actually and urgent issue of the immunity against SARS-Cov-2.

The manuscript is correctly written and its structure is good. There are only several points that need correction/clarification:

Statistical analysis should be included in the Material and method section (name of used tests, significance level, software, etc.).

There is the imprecision concerning IFN-γ concentration and anti-SARS-CoV-2 IgG expression (subheading versus first sentence of the paragraph; line 280-283)

The references should be modified strictly according to the Life style (instructions for Authors)

I recommend to add the Conclusion section, because the Discussion section is very long and complex

Reviewer 2 Report

This is a well written paper with clearly presented data. I suggest minor corrections/comments

line 59 change humorous to humoral

Figure 5. There is a decrease in B Lymphocyctes compared to the control. Do authors have a possible explanation as to why this would be?

Reviewer 3 Report

In this manuscript, the authors examined humoral and cellular immunity against SARS-CoV-2 in elderly COVID-19 convalescents. Since the COVID-19 conditions are very serious, this kind of clinical information is very valuable for the basic research as well as the clinical research. Therefore, the manuscript is suitable for the journal “Life”.

Author Response

No reviewer comments to discuss.

Round 2

Reviewer 1 Report

Thank for the revision